# Adherence to 24-hour integrated activity guidelines among infants, toddlers and preschool children in Singapore

Phaik Ling Quah[1,2]*, Benny Kai Guo Loo[3], Michael Yong Hwa Chia[4], Terence Buan Kiong Chua[4], Teresa Shu Zhen Tan[5], Poh Chong Chan[5], Kok Hian Tan[1,2]

1 Division of Obstetrics & Gynaecology, KK Women's and Children's Hospital, Singapore, Singapore, 2 Duke-NUS Medical School, Singapore, Singapore, 3 General Paediatric Service, KK Women's and Children's Hospital, Singapore, Singapore, 4 Physical Education & Sports Science, National Institute of Education, Nanyang Technological University, Singapore, Singapore, 5 Khoo Teck Puat-National University Children's Medical Institute, National University Hospital, Singapore, Singapore

* quah.phaik.ling@kkh.com.sg

**Data Availability Statement:** The data underlying the results presented in the study are available from the institute's research team at ipramho@kkh.com.sg.

## Abstract

This study examined children's adherence to the Singapore Integrated 24-Hour Activity Guidelines for Early Childhood in infants, toddlers and preschoolers aged 0–6 years. A total of 901 caregivers, comprising 219 infants, 379 toddlers, and 303 preschoolers, provided information regarding their children's physical activity (PA), screen viewing time (SVT), and sleep durations on both weekdays and weekends. Meeting the 24-hour integrated activity guidelines was defined as follows: for infants $\geq$ 30 minutes per day of tummy time or floor-based play; zero SVT; total sleep of 14–17 hours per day for ages 0–3 months, 12–16 hours per day for ages 4–11 months; for toddlers $\geq$ 180 minutes of total PA per day; zero SVT under 2 years; <1 hour for ages 2 to less than 3 years; and a total sleep of 11–14 hours per day; for preschoolers $\geq$ 180 minutes of total PA per day; SVT <1 hour per day; total sleep of 10–13 hours per day for those aged 3–5 years, and 10–11 hours per day for 6-year-olds. Chi-squared tests were used to examine the differences in guideline adherence between weekdays and weekends. Compared to weekdays, during weekends there was a higher proportion of toddlers and preschoolers adhering to the PA guidelines (68.9% vs 50.1%; 78.9% vs 55.4%, respectively, $p<0.05$), and a lower proportion of toddlers adhering to SVT (38.8% vs 21.8%; $p = 0.001$). There was a declining adherence to all three activity guidelines as age groups progressed from infants (44.7%) to toddlers (15.8%) and then to preschoolers (9.4%). Concurrently, there was a decrease in adherence to SVT recommendations across the age groups, with adherence rates being highest among infants (83.1%), followed by toddlers (15.8%), and preschoolers (9.4%). Decreasing compliance with all three guidelines, coupled with a corresponding decline in adherence to SVT guidelines as children transition from toddlerhood to preschool age, is a cause for concern. This underscores the need for proactive efforts to educate caregivers about reducing or eliminating SVT among infants and young children.

**Funding:** This study received funding from the National Medical Research Council (NMRC/CG/C008A/2017_KKH).

**Competing interests:** The authors have declared that no competing interests exist.

**Abbreviations:** NCDs, Noncommunicable diseases; WHO, World Health Organization; LPA, Light Physical Activity; EPA, Energetic Physical Activity; PA, Physical Activity; SVT, Screen viewing time; SD, Standard deviation.

## Introduction

Research has provided evidence of the association between childhood overweight and obesity with a variety of noncommunicable diseases (NCDs) [1], including type 2 diabetes, cardiovascular disease and cancers. This problem presents a significant public health concern and places a substantial strain on both healthcare and social systems [2].

The prevalence of childhood obesity has been one of the core issues of policy-makers in Singapore [3], and lifestyle behaviours have been identified as key modifiable risk factors to tackle this issue [4]. Studies have consistently demonstrated that physical activity (PA), screen-based sedentary behaviour (SB) or screen viewing time (SVT), and sleep are all individually linked to health and well-being in children [5, 6]. Higher levels of PA contribute to improved motor development, better psychosocial health, and healthier cardiometabolic function [2]. In contrast, greater SVT has been linked to higher cardiometabolic risk, unfavourable body composition and behavioural conduct, and lower self-esteem [5]. Insufficient sleep duration has been associated with higher adiposity risk, reduced emotional regulation, less successful academic achievements, and poorer quality of life [5].

Local data have indicated that children in Singapore do not meet the recommended guidelines in any of these individual lifestyle domains [7–13]. In particular, children aged 5 and below are consistently sleeping for durations that are less than the recommendations set by the United States National Sleep Foundation (NSF) for their respective age groups [7]. Furthermore, nearly all infants and toddlers aged 2 and under are already exposed to approximately 2 hours of digital media through electronic screen-based devices daily [8], while children aged 3 accumulate an average of 2.7 hours of screen time each day [10]. These figures surpass the recommended screen time limits for their respective age groups [12]. High sedentary behaviour, with preschoolers spending approximately 8.2 hours per day in this state, especially on non-school days, is primarily attributed to extended screen device usage [13]. Additionally, the 2022 Singaporean report card on physical activity for children and adolescents assigned an overall grade of "C-" for physical activity participation among Singaporean children [11], which underscores the necessity for enhancement and advancement in encouraging physical activity among Singaporean youth.

Emerging evidence suggests that health benefits are optimized when considering the interplay of physical activity, sedentary behaviour, and sleep [14]. These behaviours should not be considered in isolation, as the time spent on one behaviour is interrelated with others over the course of a 24-hour day [15]. Consequently, a unified guideline promoting a holistic "whole day matters" approach has been developed to emphasize the importance of integrating all aspects of lifestyle behaviours [15]. In Singapore, healthy lifestyle behaviours have received increasing attention among policymakers. Therefore, the Singapore Integrated 24-Hour Activity Guidelines for Early Childhood were developed and launched in 2022 [12]. These guidelines are aligned with the latest World Health Organisation (WHO) guidelines on physical activity, sedentary behaviour and sleep for children under five years of age, which were launched in 2019 [15]. The uniqueness of these guidelines compared to existing ones [16, 17] is the integration of eating behaviours and dietary choices. These are closely linked to movement behaviours, encompassing overall calorie intake and what is required to support physical activity and growth. Before the launch of these guidelines, two studies in Singapore showed that less than 10% of preschoolers met all three of the WHO 24-hour movement behaviours [9, 18]. Of the two studies, one study reported a 5.5% adherence [9], while the other reported a 9.6% adherence to all three guidelines [18]. The latter study additionally showed that the parent-reported health-related quality of life increased as the preschoolers achieved more movement behaviour recommendations [18].

To date, there has been just one comprehensive review that evaluated compliance with all three physical activity guidelines among children aged below 3 years in 11 diverse countries. The results of this review indicated that adherence rates were notably low, averaging just 13% [19]. There is a shortage of comprehensive evidence regarding the extent of adherence among children below the age of 7 to early childhood activity guidelines, especially in the case of infants and toddlers. Additionally, there is a notable absence of studies conducted in the Asian population that compare lifestyle behaviours from infancy through the preschool years. Moreover, there is a dearth of evidence concerning parental attitudes and practices related to instilling these lifestyle behaviours from home. This includes factors such as parental awareness and knowledge of guideline recommendations, as well as perceptions of their child's health. These factors, as demonstrated in previous research, are crucial in managing a child's body weight [20, 21].

As such, this study aims to describe adherence to activity guidelines for PA, sedentary SVT, and sleep, and parental knowledge, attitudes, and practices concerning child health and lifestyle behaviours in children below the age of 7 years.

## Methods

### Study sample

Caregivers participating in this study were enlisted as part of the Singaporean Lifestyle Questionnaire for infants, toddlers and preschoolers: IPRAMHO Survey on Integrated Early Childhood 24- Hour Activity for infants, toddlers and preschoolers 0–6 years old (E-24). A convenience sampling method was employed to select caregivers from various study sites in Singapore. Eligible caregivers were individuals who were either Singaporean or permanent residents, responsible for children aged 0–6 years, and possessed the ability to comprehend English. Infants were categorized as children between 0 and 11 months old, toddlers as those aged between 1 and 2 years 11 months old, and preschoolers as those between 3 and 6 years 11 months old. Before taking part, caregivers needed to provide witnessed verbal consent to the research team. Subsequently, they were given an electronic version through FormSG (https://form.gov.sg/) for completion. The research procedures for this study received formal approval for an exempt review from the SingHealth Centralized Institutional Review Board (CIRB Ref No.: 2021/2610). All the authors and caregivers of the children surveyed have provided consent for the publication of this manuscript.

### Data collection

Between October 2021and Jun 2022, a total of n = 901 survey responses were collected. The survey comprised 46 items that collected data on caregiver demographics, PA, SVT, sleep habits (nighttime sleep and daytime napping), parental practices, parental perceptions of the child's health and well-being, parental awareness and knowledge of lifestyle guidelines for their infants, toddlers and preschoolers. Caregivers self-reported demographic characteristics and reported on their child's demographics and lifestyle behaviours. They were asked to recall their children's past 7-day lifestyle and activities while completing the survey questions. The survey questionnaire was developed and reviewed by members of the Integrated 24-Hour Activity Guidelines for Early Childhood Committee. The development of the survey questionnaire went through an expert review and thorough literature review which establishes content validity. Items on PA captured the total amount of tummy time for infants, total light and energetic play for toddlers and preschoolers using previously published questions [22], while the items on sleep were adapted from the Child Sleep Habits Questionnaire [23]. Recreational

SVT was captured as the average time a child spends on a screen [10]. The survey took 10–15 minutes in total duration.

## Assessment of physical activity

Caregivers were asked to report the average duration per day of the child's tummy time or time spent in floor-based play on a typical weekday or weekend if the child was 0–11 months old [24]. For toddlers and preschoolers, light physical activity (LPA) and energetic physical activity (EPA) during the weekday and weekend were assessed using questions: "How much time did your child spend in light physical activity (Total per day on average)?" and "How much time did your child spend in energetic physical activity (Total per day on average)? ". LPA was defined as: "Low energy activities such as walking at a leisurely or moderate pace, biking/cycling or jumping at an easy pace and swimming with support of an adult, playing with a sibling or an adult indoors, dancing to music or attending a music and movement class", and EPA was defined as: "Activities that make your child breathe faster/harder than normal, (e.g. skipping, hopping, swimming and dancing with moderate to hard effort, riding a scooter/balance bike/cycling at the moderate to hard speed, attending a structured indoor class like Kinder Gym or My Gym or any other indoor sports)." Total PA for toddlers and preschoolers were the sum of LPA and EPA duration (in minutes) separately, on weekdays and weekends based on the recommended guidelines to meet 180 minutes or more of LPA and EPA combined per day [12].

## Assessment of screen time

SVT duration (in minutes) during the weekday and weekend was assessed using the questions "How much time did your child spend on recreational screen time (Total per day on average)?" SVT captured did not include home-based learning, but included the following screen devices: televisions (viewing or playing television games); computers; and handheld devices (e.g., video games and hand phones including tablets for watching videos) [10].

## Assessment of sleep

Caregivers reported their child's total amount of nighttime sleep and naptime duration (in hours and minutes) separately for weekdays and weekends. Nighttime sleep duration was captured with the questions adapted from the Child's Sleep Habit Questionnaire [23]: "Child's usual amount of sleep each night (Total per night on average, not including daytime naps)". Naptime duration was captured with the question: "What is the total average time spent on nap(s) (Total per day on average)?" Total sleep time was calculated by taking the sum of night and daytime naps.

## Parental practices

Parental practices with their infants, toddlers and preschoolers were assessed with seven questions: (1) "I am active with, or in front of my child"; (2) "My child is restrained daily from free spontaneous movement"; (3) "My child has a consistent bedtime routine and bedtime"; (4) "I limit my own recreational screen time use when I am spending time with my child"; (5) "I prepare or provide meals for my child that are well-balanced and generally healthy"; (6) "I limit my own intake of unhealthy food and beverages"; and (7) "My child has at least one meal with the family (refers to family members or relatives living under the same roof) each day". The response options were on a 5-point Likert Scale of "Never", "Rarely", "Occasionally", "Frequently" and "All the time".

## Parental perception of child's health

Parental perception of their child's weight, calorie intake, PA, SVT exposure and sleep were assessed with the questions previously published [25] "Do you think your child is . . .", with the response options of "Overweight", "Normal weight", "Underweight" and "Not sure". The other 3 questions were (1) "Do you feel that your child receives an adequate amount of physical activity to benefit his/her growth, development and health?"; (2) "Are you concerned about the amount of recreational screen time your child is currently exposed to?"; and (3) "Do you think your child is getting adequate sleep to support his/her growth, development and health?", with response options of "Yes", "No" and "Not sure".

## Parental awareness and knowledge of guideline recommendations

To assess parental awareness of available lifestyle guidelines for children, caregivers were asked 3 separate questions regarding PA, sleep and SVT: "Are you aware that there are guidelines on the amount of physical activity your child should be receiving per day as a toddler/preschooler, and the limit on the restraint time for your infant?"; "Are you aware that there are guidelines on the amount of sleep your child should be receiving per day as an infant/toddler/preschooler?", and "Are you aware that there are guidelines on the amount of screen time your child should be receiving per day as an infant/toddler/preschooler?" with the response options of "Not aware" and "Aware but not practising" and "Aware and practising". In addition, caregivers were also asked to estimate what they think is the current guideline recommendation for PA, sleep and SVT with questions: (1) "What do you think is the current guideline of the amount of time toddlers and preschoolers should spend in a variety of physical activities at any intensity throughout the day? For infants, what do you think is the guideline for amount of time in tummy-time or floor-based play?" (2) "What do you think is the guideline for the total amount of sleep your infant/toddler/preschooler requires per day?" (3) "What do you think is the guideline for recreational screen viewing time allowance (maximum limit) for your infant/toddler/preschooler per day?". Knowledge of guideline recommendations was assessed by questions, and based on their answers, caregivers were categorized into two groups: accurate and inaccurate.

## Meeting guideline recommendations for infants, toddlers and preschoolers

Infants were classified as meeting physical activity guidelines if they met ≥30 minutes/day of tummy time or floor-based play. Toddlers and preschoolers were classified as meeting physical activity guidelines with total PA of ≥180 minutes/day. Infants and toddlers younger than 2 years were classified as meeting the SVT guidelines if they engaged in no screen time, while toddlers between 2 years and younger than 3 years, and preschoolers between 3 years and 6 years 11 months met the SVT guidelines if they engaged in <1 hour/day. Children were classified as meeting sleep guidelines if they slept a daily total amount of 14–17 hours (for 0–3 months of age) and 12–16 hours (for 4–11 months of age), 11–14 hours (1–2 years of age), 10–13 hours (for 3–5 years of age) and 10–11 hours (for 6 years of age). Guideline adherence was in-line with the Singapore Integrated 24-Hour Guidelines for early childhood [12].

## Statistical analyses

Continuous datasets that were normally distributed are presented as the mean and standard deviation (SD), while nonnormally distributed datasets are presented as the median and interquartile range (IQR). The Mann-Whitney U-test for nonnormally distributed datasets, T-Tests for normally distributed datasets, and the chi-square test were used to compare continuous

and categorical variables, respectively, between weekdays and weekends. Frequencies and percentages were used to describe the proportion of children meeting the movement behaviour guidelines. Sex- and age-specific body mass index (BMI) z-scores were derived using the WHO references for children 0–5 years of age [26] and children 5–6 years of age [27]. Statistically significant results were determined at 2-sided $p < 0.05$. All analyses were performed using STATA software version 13 (StataCorp, College Station, US).

## Results and discussion

### Caregiver characteristics

A total of 901 caregivers participated in the E-24 study and caregiver characteristics are presented in Table 1. In summary, 70.9% of the caregivers were mothers, of Chinese ethnicity (64.2%), with university-level education (66.5%), and 56.4% were between 30 and 35 years old. Of all the children, 69.3% were first-born, approximately half were males (51.3%), 24.3% were infants (0–11 months), 42.1% were toddlers (1 year- 2 years 11 months) and 33.6% were pre-schoolers (3 years– 6 years 11 months). On average, infants were 5.8 months old (SD 3.3),

**Table 1. Characteristics of the participants in the E24 study (N = 901).**

| Caregiver characteristics | N | % |
|---|---|---|
| Chinese | 578 | 64.2 |
| Malay | 222 | 24.7 |
| Indian | 83 | 9.2 |
| Others | 17 | 1.9 |
| **Relationship with child** | | |
| Mother | 639 | 70.9 |
| Father | 253 | 28.1 |
| Grandparent | 7 | 0.8 |
| Others | 2 | 0.2 |
| **Birth order** | | |
| First-born | 624 | 69.3 |
| Second-or-later-born | 276 | 30.7 |
| **Highest education** | | |
| University | 599 | 66.5 |
| Post-secondary | 278 | 30.9 |
| Primary and secondary | 24 | 2.6 |
| **Age** | | |
| <30 | 138 | 15.4 |
| 30–35 | 505 | 56.4 |
| >35 | 252 | 28.2 |
| **Child characteristics** | | |
| **Child sex** | | |
| Male | 461 | 51.3 |
| Female | 437 | 48.7 |
| **Child age group** | | |
| Infant (0–11 months) | 219 | 24.3 |
| Toddler (1 year-2 years 11 months) | 379 | 42.1 |
| Pre-schooler (3 years—6 years 11 months) | 303 | 33.6 |

Missing data: Ethnicity (n = 1), age (n = 6)

**Table 2. Characteristics of the infants, toddlers and pre-schoolers in the E24 study.**

| | Infant (n = 219) | Toddler (n = 379) | Pre-schooler (n = 303) |
|---|---|---|---|
| | Mean (SD) | Mean (SD) | Mean (SD) |
| Child age (months) | 5.8 (3.3) | 24.6 (6.5) | 55.0 (13.0) |
| Weight (z-score) | 0.06 (1.3) | 0.73 (1.4) | -0.05 (1.04) |
| Length (z-score) | 0.5 (1.4) | 0.29 (1.3) | 0.52 (1.3) |
| BMI (z-score) | 0.5 (1.4) | 0.4 (1.3) | 0.5 (1.3) |
| **Childcare attendance, n (%)** | | | |
| Yes | 25 (11.4) | 144 (38.2) | 173 (58.5) |
| No | 194 (88.6) | 233 (61.8) | 123 (41.6) |
| **Chronic illness, n (%)** | | | |
| Yes | 0 (0) | 5 (1.3) | 7 (2.5) |
| No | 219 (100) | 368 (98.7) | 269 (97.5) |

Characteristics of the infants, toddlers and pre-schoolers were presented as mean ± SD unless stated otherwise.

Missing data: Child age—Infant (n = 7); toddler (n = 12); pre-schooler (n = 9); childcare attendance–toddler (n = 2), pre-schooler (n = 7); chronic illness–Toddler (n = 1), Pre-school (n = 70).

Out of range values: weight z-scores–Infant (n = 39); toddler (n = 65); pre-schooler (n = 101), length z-scores–Infant (n = 103); toddler (n = 189); pre-schooler (n = 132), BMI z-scores–Infant (n = 100); Toddlers (n = 200), pre-schooler (n = 137). childcare attendance–toddler (n = 2), pre-schooler (n = 7); chronic illness–Toddler (n = 6), Pre-school (n = 27).

toddlers were 24.6 months old (SD 6.5) and the preschoolers were 55.0 months old (SD 13), with BMI Z-scores of 0.5 (SD 1.4), 0.4 (SD 1.3) and 0.5 (SD 1.3), respectively. The proportion of children attending childcare was 11.4% of infants, 38.2% of toddlers and 58.5% of preschoolers. Almost all the children in this study were free from any chronic illnesses (>97%) (Table 2).

## Caregiver-reported physical activity, sedentary behaviour and sleep durations

Table 3 describes the caregiver-reported PA, SVT and sleep of their infants, toddlers and preschoolers. Infants engaged in a median tummy time of 45 (IQR 30–120) minutes/day in a week, whereas toddlers and preschoolers had higher total PA duration during weekends, compared to weekdays ($p<0.001$). There was higher SVT during weekends than on weekdays across all age groups ($p<0.001$). In only the preschoolers, total sleep durations were higher on weekends, than on weekdays [mean 11.2 (SD1.5) vs 10.9 (1.3) hours/day, $p<0.001$].

## Adherence to guideline recommendations for physical activity, sedentary behaviour and sleep

Table 4 describes the proportion of children meeting the PA, SVT and sleep guidelines according to age group. In infants, the proportions meeting tummy time (76.8%), SVT (83.0%) and sleep (72.0%) guidelines did not differ between weekdays and weekends ($p>0.05$). A higher proportion of infants met all three guidelines during weekdays (45.2% vs 44.2%, $p<0.001$).

On a weekend compared to a weekday, a higher proportion of toddlers met PA guidelines (68.9% vs 50.1%, $p<0.001$); conversely, a lower proportion met SVT guidelines (21.8% vs 38.8%, $p<0.001$), and there were no differences in meeting sleep guidelines (79.1% vs 82.6%, $p>0.05$). A higher proportion of toddlers met all three guidelines on weekdays, than on weekends (18.3% vs 13.2%, $p<0.001$).

**Table 3. Physical activity, screen viewing time and sleep on weekdays and weekends by age group.**

| | Infants (n = 219) | | | Toddlers (n = 379) | | | Pre-schoolers (n = 303) | | |
|---|---|---|---|---|---|---|---|---|---|
| | **Weekday** | **Weekend** | **p value** | **Weekday** | **Weekend** | **p value** | **Weekday** | **Weekend** | **p value** |
| | **Median (IQR)** | **Median (IQR)** | | **Median (IQR)** | **Median (IQR)** | | **Median (IQR)** | **Median (IQR)** | |
| **#Physical activity (mins/day)** | | | | | | | | | |
| Tummy time | 45 (30–120) | | - | - | - | | | | |
| Light activity | - | | - | 120 (60–180) | 120 (90–240) | 0.001 | 90 (60–120) | 120 (60–180) | 0.001 |
| Energetic activity | - | | - | 30 (15–60) | 60 (30–120) | 0.001 | 60 (30–120) | 120 (60–180) | 0.001 |
| **Total physical activity** | | | | 180 (90–240) | 210 (135–360) | 0.001 | 180 (120–240) | 240 (180–300) | 0.001 |
| **Sedentary behaviour** | | | | | | | | | |
| Screen viewing time (mins/day) | 30(30–60) * | 60 (30–60) * | 0.001 | 40 (25–60) | 60 (30–120) | 0.001 | 60 (60–120) | 120 (60–180) | 0.001 |
| **~Sleep (hours/day)** | | | | | | | | | |
| Night time sleep, mean (SD) | 11.4 (3.3) | 11.1 (2.1) | 0.10 | 11.0 (1.5) | 10.9 (1.4) | 0.04 | 9.5 (1.3) | 9.9 (1.4) | 0.001 |
| Day time sleep, mean (SD) | 3.0 (2.3) | 3.1 (2.4) | 0.05 | 1.8 (1.5) | 2.0 (1.4) | 0.006 | 1.5 (1.4) | 1.4 (1.5) | 0.007 |
| Total sleep, mean (SD) | 14.0 (2.0) | 14.0 (2.2) | 0.98 | 12.8 (1.5) | 12.9 (1.6) | 0.19 | 10.9 (1.3) | 11.2 (1.5) | 0.001 |

Data was captured for weekdays and weekends except for infant tummy time. Missing data: tummy time (n = 8), missing screen time for toddlers (n = 7) on weekdays, and n = 8 on weekends, missing screen time for pre-schoolers (n = 5) for both weekday and weekend.

*Median was calculated only for the 37 infants with any SVT

# Any implausible physical activity duration for toddlers and pre-schoolers above 720 minutes was removed from the analysis: (Toddlers = n = 14 for weekdays, n = 15 for weekends; Pre-schoolers physical activity n = 5 for weekday, n = 8 for weekend)

~ Any implausible sleep duration <6 hours or >20 hours in total were removed from the analysis (Infants = 14 for both weekday and weekend, Toddlers n = 11 for weekday and weekend, Pre-schoolers n = 13 on weekday and weekends)

On weekends compared to weekdays, a higher proportion of preschoolers met the PA guidelines (78.9% vs 55.4%, $p<0.001$), but there were no differences in meeting SVT (21.5% vs 21.5%, $p>0.05$) or sleep guidelines (82.5% vs 80.7%, $p>0.05$). In contrast, for infants and toddlers, a higher proportion of preschoolers met all three guidelines during weekends compared to weekdays (10.4% vs 8.3%, $p<0.001$).

## Parenting practices, perceptions, awareness and knowledge

S1 Table describes parenting practices across all age groups. Caregivers of preschoolers exhibited the lowest likelihood of frequently engaging in physical activities with their children or restricting their children from spontaneous movement ($p<0.001$). Moreover, caregivers of preschoolers were also the most likely to occasionally restrict their own screen time and their consumption of unhealthy food and beverages ($p<0.001$). On the other hand, caregivers of toddlers were more inclined to provide well-balanced and generally healthy meals to their children, while caregivers of preschoolers were most likely to have at least one daily family meal together, frequently ($p<0.001$) (S1 Table). S2 Table provides insights into parental perceptions of their child's health. Caregivers of toddlers were more inclined to perceive their child as being underweight and receiving sufficient physical activity, whereas caregivers of preschoolers expressed greater concerns about their child's SVT and were less likely to believe their child was getting enough sleep ($p<0.01$) (S2 Table).

S3 Table, on the other hand, describes parental awareness of guideline recommendations and their knowledge of specific activity guideline recommendations. Interestingly, caregivers of infants were most likely to think they were aware of and adhering to sleep and SVT guidelines, but only 58.9% and 27.1% of them, respectively, were accurate in their knowledge. Caregivers of toddlers were more prone to thinking they were aware of and complying with

**Table 4. Meeting guidelines for physical activity, screen viewing time and sleep on weekdays and weekends by age group.**

| | Infants (n = 219) | | |
| --- | --- | --- | --- |
| | **Weekday** | **Weekend** | **p value** |
| | **n (%)** | **n (%)** | |
| **Physical activity** | | | |
| Tummy time | 162 (76.8) | | |
| **Sedentary behaviour** | | | 1.000 |
| Screen viewing time | 182 (83.1) | 182 (83.1) | |
| **Sleep (hours/day)** | | | 0.84 |
| Only infants 0–3 months old (n = 79) | 34 (43.0) | 36 (45.6) | |
| Only infants 4–11 months old (n = 140) | 114 (81.4) | 111 (79.3) | |
| All infants 0–11 months old | 148 (72.2) | 147 (71.7) | |
| **Meeting all three guidelines** | 89 (45.2) | 87 (44.2) | 0.001 |
| | Toddlers (n = 379) | | |
| | **Weekday** | **Weekend** | **p value** |
| | **n (%)** | **n (%)** | |
| **Physical activity** | | | 0.001 |
| Total light and energetic play | 183 (50.1) | 251 (68.9) | |
| **Sedentary behaviour** | | | 0.001 |
| Screen viewing time | 144 (38.8) | 81 (21.8) | |
| **Sleep** | | | 0.11 |
| | 304 (82.6) | 291 (79.1) | |
| **Meeting all 3 guidelines** | 66 (18.3) | 47 (13.2) | 0.001 |
| | Pre-schoolers (n = 303) | | |
| | **Weekday** | **Weekend** | **p value** |
| | **n (%)** | **n (%)** | |
| **Physical activity** | | | |
| Total light and energetic play | 165 (55.4) | 233 (78.9) | 0.001 |
| **Sedentary behaviour** | | | |
| Screen viewing time | 64 (21.5) | 64 (21.5) | 1.000 |
| **Sleep** | | | 0.58 |
| | 231 (82.5) | 230 (80.7) | |
| **Meeting all 3 guidelines** | 23 (8.3) | 29 (10.4) | 0.001 |

Missing data: Infant sleep (n = 14), toddlers' physical activity (n = 14 for weekdays, n = 15 for weekends), toddlers screen time (n = 7 on weekdays and n = 8 on weekends), toddlers sleep (n = 11), pre-schoolers physical activity (n = 5 for weekday, n = 8 for weekend), pre-schoolers SVT n = 5 for weekday and weekend, pre-schooler sleep n = 23 for weekday and n = 18 for weekend.

physical activity (PA) guidelines, although only 38.4% had accurate knowledge in this regard (S3 Table).

Our study addressed the existing knowledge gap regarding adherence to 24-hour activity guidelines in a culturally diverse Asian population within Singapore, focusing on infants, toddlers and preschool children below 7 years of age. Notably, two significant findings emerged from this study. First, there was a decline in the percentage of children who met all three criteria for PA, SVT, and sleep as they transitioned from infancy (~45%), toddlerhood (~18%) to preschool age (~10%). Second, while a larger proportion of infants and toddlers adhered to all three recommendations on weekdays, the opposite trend was observed among preschoolers.

The adherence to the 24-hour activity guidelines in this sample of infants (mean age:5.6 months) is higher (~45%) than what has been reported in other studies. For instance, in a study involving Australian infants with an average age of 3.6 months, only 4% met all the overall guidelines, which included the addition of meeting infant restraint guidelines [28]. Similarly, in a study involving Canadian infants, the percentages meeting all three guidelines were 12.4%, 18.0%, and 27.6% at 2 months, 4 months, and 6 months of age, respectively [29]. It is noteworthy that the primary distinction between our study and the previously mentioned studies was the notably higher compliance with the recommendation of zero screen time, which was approximately 83% in our sample, in contrast to the lower percentages reported in Australian infants (27.9% at age 3.6 months) [28] and Canadian infants (37.2% at age 6 months) [29]. In this study, infants were found to have a daily screen time exposure ranging from a median of 0.5 to 1 hour around the age of 6 months. A local study indicated that infants, by the age of 12 months, had a higher daily screen time exposure averaging 2.01 hours [30]. Interestingly, a lower proportion of infants between ages 0–3 months old (~45%) met sleep recommendations compared to infants between ages 4–11 months old (~72%). These findings concur with data from the Growing Up in Singapore Toward Healthy Outcomes (GUSTO) study where only 30% of infants at 3 months old met sleep recommendations, compared to 47% at 6 months and 56% at 9 months [31]. The average sleep duration in infants from this study was also similar to that in previously published local studies [32, 33]. A study in Canadian infants displayed a similar trend of higher parent-reported adherence to sleep guidelines in 4 (73.6%) and 6 (80.4%) months old, than in 2-month-olds (49.2%). These findings could imply that sleep-wake patterns during the early stages of infancy tend to be unpredictable until infants develop day-night (circadian) rhythms [34]. This unpredictability might lead to reduced adherence to sleep recommendations during the initial months or make it challenging for caregivers to accurately assess the overall duration of their infants' sleep.

Among toddlers with an average age of 2 years old, we observed a decline in overall adherence to activity guidelines compared to infants, possibly attributed to the reduced adherence to physical activity and screen time guidelines. Overall, the trends observed in this study align with findings from an Australian study in toddlers with an average age of 26.9 months where higher proportions adhered to physical activity (81.6%) and sleep (82.3%), while the lowest adherence was noted in screen time use (30.5%) [35]. So far, toddlers' compliance with 24-hour activity guidelines have only been researched in three Western populations. The proportions conforming to all guidelines exhibited minimal variation, aligning closely with our established ranges of 13.2% on weekdays and 18.3% on weekends. Notably, Australia reporting a proportion of 8.9% in the lower end [36] and 20.1% on the higher end [35], while Canada observed 11.9% meeting all three guidelines [37]. Local evidence has indicated that children between 12 and 24 months old typically accumulate an average of 2 hours of screen time daily [10]. As they progress into toddlerhood (18 to 24 months), approximately 90% of children engage in daily passive screen viewing [10]. Our findings on screen time adherence concur with studies undertaken in other countries [38, 39]. What is intriguing is that our study revealed a significant increase in screen time viewing (up to 2 hours per day) during weekends, due to children spending more time at home, as opposed to weekdays when nearly a quarter of children attend childcare centres. This also explains the higher adherence to screen time guidelines during the weekdays, as screen time use in childcare is supervised. Furthermore, it is postulated that engagement in higher durations of screen time might displace engagement in physical activity [40]. However, the lower engagement in physical activity observed from our study during the weekdays despite the lower engagement in screen time use could be due to the inability of caregivers to gauge total physical activity per day in children who were attending childcare. The average sleep duration of approximately 12.9 hours/day among toddlers

concurs with other studies conducted in Singapore children aged 2 years old [33, 41]. The lower nighttime sleep duration on weekdays could be explained by the longer daytime nap durations, possibly contributed by compulsory naptime practices in all childcare centres in Singapore [41]. A study published in Japanese children reported similar findings in the association between longer afternoon naps and shorter nighttime sleep-in toddlers as young as 1.5 years old [42].

At 4.5 years of age, the collective adherence to the three activity guidelines decreased to a range of 8.3% to 10.4% among preschoolers. This aligns with findings from a study involving Singaporean children aged 5.5 years old, where only 5.5% were observed to meet all three guidelines [9]. In densely populated Asian cities like Hong Kong and Japan, only 2.9% [43] and 7.2% [44] respectively, of preschoolers were reported to comply with all three guidelines, a proportion lower than the rates observed in Singapore [45]. In Western populations, two studies revealed higher percentages of children adhering to all three guidelines [45, 46], while one observed very low rates of adherence [47]. In a study conducted in rural Brazil, 10% of children aged between 3–6 years met these criteria which concurred with our study observations [46]. In Australia, the proportion was slightly higher at 14.9% [45], whereas a study in the United States reported a percentage of 1.6% [47]. In our study, overall reduction in adherence to all three guidelines was accompanied by a further reduction in the adherence to screen time especially on weekdays to only 21.8%, while the adherence to physical activity and sleep guidelines remained relatively similar to that of toddlers. Again, similar to our findings, the study from Brazil has observed that approximately half of the children were meeting physical activity guidelines, while only approximately 20% were meeting screentime guidelines [46]. The trend of lower physical activity engagement during the weekdays concurs with the observations from the study conducted in Hong Kong [43], but contrasts with another study conducted in Singapore preschoolers, which reported higher engagements on weekdays, whether it was parent-reported data or objectively measured using wrist-worn accelerometers [13]. The study collected parent-reported data, which might not be as accurate as objectively measured data. Additionally, the variability in childcare arrangements for children at this age—some not in childcare, some in full-time childcare, and others spending half a day in childcare with the rest of the day with noncaregivers (helpers, or grandcaregivers)—could make it challenging to accurately gauge the amount of physical activity engagement spread throughout the day [48]. Preschoolers engaged in as much as 3 hours per day to screen activities on weekends. This pattern of escalating screen time as children age mirrors observations made in a local study involving children aged 6 to 24 months [32], and aligns with findings from a study in the United States concerning children aged 0 to 8 years [49]. Our nighttime sleep duration in this age group is consistent with findings from other local studies [33, 41]. At this stage, a notable disparity emerged in the total sleep duration children were obtaining, with greater amounts during weekends (11.2 hours per night) in contrast to weekdays (10.9 hours per night), primarily due to extended nighttime sleep. It is worth highlighting that nearly 60% of children had commenced either full-day or half-day childcare by this age. Compulsory naptimes at schools would explain the longer daytime naps during the weekdays, which might have been attributed to shorter weekday nighttime sleep. Thus, longer day sleep might lead to later bedtime, shorter nocturnal sleep and possibly, poorer night sleep quality [50]. A study conducted with children aged 50–72 months yielded comparable findings. It noted that children who were required to nap for more than 60 minutes during childcare had notably less nighttime sleep in comparison to those who had nap durations of 0–60 minutes [51].

We analysed data on parenting attitudes, practices, and awareness of guidelines, particularly focusing on variations among infants, toddlers, and preschool-age children. Interestingly, we observed a decline in the proportion of caregivers frequently engaging in physical activities

and preparing healthy foods from infancy to preschool age. At preschool age, half of the caregivers did not think their child was receiving adequate physical activity or were not sure, and one tenth were concerned about their children being underweight. Our findings concurred with a study that showed that caregivers of children aged 5–7 years believed that promoting physical activity was not necessary, and that their children were sufficiently active [52]. Conversely, the proportion of caregivers who would only occasionally limit their own screen time or discretionary food intake was highest in the preschooler age group. At this age, half of the caregivers expressed concerns about their child's excessive screen time. A local study has shown that at least 70% of caregivers of preschool-aged children expressed concern for child digital media use [53]. Caregivers are key contributors to shaping a child's lifestyle behaviour [54, 55], specifically caregivers influence children by modelling healthy behaviours and providing an environment necessary for the child's growth and development. Children tend to engage in more physical activity when caregivers are supportive and encouraging of such behaviours [56], and a study has shown that a child is 5.8 times more likely to be involved in physical activity if caregivers actively participate as well [57]. Screen time viewing in children below 7 years of age [58], and as young as 2 years of age [10] are found to be highly correlated with parental practices. In Singapore, a quarter of toddlers and over half of preschoolers received childcare from sources outside the home, and a local study has highlighted the impact of diverse childcare arrangements on children's health behaviours. Notably, children who spend more time with nonparent caregivers, such as domestic helpers or grandcaregivers, exhibit increased adiposity and engage in higher levels of sedentary behaviours, including extended screen time, compared to children who were placed in an all-day childcare [48].

Recognizing a caregiver's understanding of child development as a crucial aspect of caregiver support is a widely recognized central theme. Acquiring specific knowledge is a fundamental caregiver skill. In our study, we noted that, concerning PA, caregivers who were aware about the guidelines but opted not to implement them were most prevalent among preschoolers. Additionally, accuracy in knowledge of PA guidelines was lowest within this age group. Similarly, caregivers who were aware of SVT guidelines, but chose not to adhere to them were most common among preschoolers. Interventions aimed at improving parents' understanding, abilities, and social support related to infant feeding, diet, physical activity, and television watching, have been shown to be related to a reduced likelihood of unfavourable behaviours. These included the consumption of sugary snacks and the amount of time spent watching television [59]. In older age groups of children, caregiver practices seemed to exhibit fewer positive trends. This could be attributed to a decrease in their level of knowledge regarding lifestyle recommendations for their child, possibly because they tend to rely more on childcare centers to oversee their child's education, dietary choices, and physical activity. At least 50% of all children attending childcare by the age of 5 were there for at least five full-days a week [48]. Nonetheless, prioritizing parental engagement and involvement with children remains essential. This is because, despite the decreased fast-food and deep-fried food intake that has been associated with childcare attendance in Singapore [48], there were no associations with increased intake of fruits or vegetables. Moreover, overall physical activity engagement remains low in preschoolers even across school and non-school days [13].

The strengths of this study include the comprehensive data collected on all movement behaviours, and the collection of data across three different age groups from 0 years to 6 years of age. This allowed for the comparison of these lifestyle behaviours, including parenting knowledge, attitudes and practices of lifestyle recommendations for infants, toddlers and preschoolers. This study also has limitations, that need to be addressed. These include the subjective self-reported responses of caregivers in the survey about their children's activities, which may not be as accurate as objective measurements using motion sensors. Another constraint

lies in the adoption of the convenience sampling method in this research, potentially leading to the sample's lack of representativeness when compared to the entire population of Singapore. Additionally, the sample was made up mostly of caregivers with at least a university-level degree and might not be reflective of caregivers with other educational qualifications.

## Conclusions

In conclusion, adherence to all three activity guidelines varied across age groups, showing a notable decline from infancy to preschool age. This decline was largely attributed to lower proportions of toddlers and preschoolers adhering to screen time guidelines compared to infants. Future interventions to address childhood obesity may benefit from targeting integrated movement-related behaviours, suggesting the need for strategies in Singapore to commence early, potentially through parenting education during pregnancy, to prevent the gradual decline in adherence across the years.

## Supporting information

**S1 Table. Parental practices of lifestyle behaviours by age group.**
(DOCX)

**S2 Table. Parental perceptions of child's health by age group.**
(DOCX)

**S3 Table. Parental awareness of lifestyle behaviour guideline recommendations by age group.**
(DOCX)

## Acknowledgments

Integrated 24-Hour Activity Guidelines for Early Childhood Committee: Benny Kai Guo LOO, KK Women's and Children's Hospital, Singapore; Dinesh SIRISENA, Khoo Teck Puat Hospital, Singapore; Falk MUELLERRIEMENSCHNEIDER, National University of Singapore, Singapore; Michael Yong Hwa CHIA, Nanyang Technological University, Singapore; Benedict Chi'-Loong TAN, Changi General Hospital, Singapore; Ngiap Chuan TAN, SingHealth Polyclinics, Singapore; Professor Oon Hoe TEOH, KK Women's and Children's Hospital, Singapore; Ethel Jie Kai LIM, KK Women's and Children's Hospital, Singapore; Mohammad Ashik ZAINUDDIN, KK Women's and Children's Hospital, Singapore; Joanne Shumin GAO, KK Women's and Children's Hospital, Singapore; Poh Chong CHAN, National University Hospital, Singapore; Teresa Shu Zhen TAN, National University Hospital, Singapore; Nirmal KAVALLOOR VISRUTHAN, KK Women's and Children's Hospital, Singapore; Victor Samuel RAJADURAI, KK Women's and Children's Hospital, Singapore; Moira Suyin CHIA, KK Women's and Children's Hospital, Singapore; Nur Adila Binte Ahmad HATIB, KK Women's and Children's Hospital, Singapore; Shirong CAI, Singapore Institute for Clinical Sciences Agency for Science, Technology and Research (A*STAR), Singapore; Ju Lynn ONG, National University of Singapore, Singapore; Professor June Chi Yan LO, National University of Singapore, Singapore; Mary Foong Fong CHONG, National University of Singapore, Singapore; Le Ye LEE, National University Hospital, Singapore; Elaine Chu Shan CHEW, KK Women's and Children's Hospital, Singapore; Ratnaporn SIRIAMORNSARP, SingHealth Polyclinics, Singapore; Miriam LEE, Sport Singapore, Singapore; Aaron SIM, Health Promotion Board, Singapore; Jean ANG, KK Women's and Children's Hospital, Singapore; Phaik

Ling QUAH, KK Women's and Children's Hospital, Singapore; Kok Hian TAN, KK Women's and Children's Hospital, Singapore.

## Author Contributions

**Conceptualization:** Benny Kai Guo Loo, Kok Hian Tan.

**Data curation:** Phaik Ling Quah, Michael Yong Hwa Chia, Terence Buan Kiong Chua, Teresa Shu Zhen Tan, Poh Chong Chan.

**Formal analysis:** Phaik Ling Quah.

**Methodology:** Phaik Ling Quah.

**Project administration:** Phaik Ling Quah.

**Supervision:** Phaik Ling Quah.

**Writing – original draft:** Phaik Ling Quah.

**Writing – review & editing:** Phaik Ling Quah, Benny Kai Guo Loo, Michael Yong Hwa Chia, Terence Buan Kiong Chua, Teresa Shu Zhen Tan, Poh Chong Chan, Kok Hian Tan.

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
