## [Decision Letter · Decision Letter 0]

14 Jan 2024

PONE-D-23-34707Adherence to 24-hour Integrated Activity Guidelines among Infants, Toddlers and Preschool children in Singapore.PLOS ONE

Dear Dr. Quah,

Thank you for submitting your manuscript to PLOS ONE. After careful consideration, we feel that it has merit but does not fully meet PLOS ONE’s publication criteria as it currently stands. Therefore, we invite you to submit a revised version of the manuscript that addresses the points raised during the review process.

We look forward to receiving your revised manuscript.

Kind regards,

Stevo Popovic, Ph.D.

Academic Editor

PLOS ONE

4. In the online submission form you indicate that your data is not available for proprietary reasons and have provided a contact point for accessing this data. Please note that your current contact point is a co-author on this manuscript. According to our Data Policy, the contact point must not be an author on the manuscript and must be an institutional contact, ideally not an individual. Please revise your data statement to a non-author institutional point of contact, such as a data access or ethics committee, and send this to us via return email. Please also include contact information for the third party organization, and please include the full citation of where the data can be found.

Additional Editor Comments:

Dear Author(s),

I am so please to inform you I have collected three reviews and we are ready to go ahead with the evaluation process. It is your turn now to read the reviews and carefully revise the manuscript according to the requirements of the reviewers. I have to highlight that there are two positive reviews and one negative, so I would appreciate if you prepare the revision of your manuscript in line of two positive reviews but also to consider the negative one and accept all the comments you believe that can help improving the manuscript; however, I would appreciate if you prepare the comments with some arguments and adequate justifications for the rest and try to persuade the reviewer to change his/her mind.

Sincerely.

Reviewers' comments:

Reviewer's Responses to Questions

**Comments to the Author**

1. Is the manuscript technically sound, and do the data support the conclusions?

Reviewer #1: Yes

Reviewer #2: Yes

Reviewer #3: Yes

2. Has the statistical analysis been performed appropriately and rigorously? 

Reviewer #1: Yes

Reviewer #2: Yes

Reviewer #3: Yes

3. Have the authors made all data underlying the findings in their manuscript fully available?

Reviewer #1: Yes

Reviewer #2: Yes

Reviewer #3: Yes

4. Is the manuscript presented in an intelligible fashion and written in standard English?

Reviewer #1: Yes

Reviewer #2: Yes

Reviewer #3: Yes

5. Review Comments to the Author

Reviewer #1: I believe this is an interesting study and there is a merit for this journal. Physical activity is an essential need for children. Unfortunately, sedentary life style and screen time are the most two important problems now. Authors made such a good systematic data collection and the date presented very well. However, reliability and validity of the tests and measurement should be reported. In addition, more up to date literature review (2020 and above) should be used in discussion section and references. In addition, future suggestions and practical applications should be added for future research studies. Finally, I wonder if authors checked any thing about gender and socio economic status of children or family income. These things could be important variables as well. Thank you.

Reviewer #2: Did that drop lead to worse results on some tests?

Or maybe to a worse state of health?

These would be useful answers on the basis of which it would be known whether it is justified to adhere to the 24-Hour Activity Guidelines.

This way the manuscript looks just like an overview newspaper article.

It has no scientific value.

By working with a control group, the manuscript would gain a completely new dimension. In this way, without a control group it is not scientific. It does not offer proofs and solutions.

I cannot recommend it for publication.

I think that it does not reach the quality level of the Plos One journal.

Reviewer #3: I read the study carefully and liked it very much. Methodologically, it is written according to the rules, I have only one dilemma that I leave to the editor to solve. Namely, in tables 1 and 2 there are missing data for a certain number of respondents (some of whom the author listed below the table, and some he failed to list), so the question arises whether some respondents who do not have complete data should be included in the study? Maybe the final results in some variables would be different? And on line 304, the data 71.9% do not agree with the data from table 4 (79.1%)

6. PLOS authors have the option to publish the peer review history of their article (what does this mean?). If published, this will include your full peer review and any attached files.

Reviewer #1: **Yes: **Ferman Konukman

Reviewer #2: No

Reviewer #3: **Yes: **Jovan Gardasevic

---

## [Author Response · Author response to Decision Letter 0]

22 Jan 2024

Editor-in-Chief and reviewers

PLOS ONE

17th January 2024

Dear Editor and reviewers,

Re: Submission of revised manuscript for evaluation and responses to reviewers’ comments and suggestions.

On behalf of the co-authors, we would like to thank PLOS ONE for reviewing this manuscript and the opportunity to re-submit a revised version of the manuscript entitled “"Adherence to 24-hour Integrated Activity Guidelines among Infants, Toddlers and Preschool children in Singapore” for consideration in PLOS ONE. We greatly appreciate the constructive comments and suggestions provided by the reviewers which have helped improve the quality of the manuscript.: 

Reply to journal requirements:

Reply: The revised manuscript has been formatted to the meet the PLOS ONE style requirements, including those for file naming.

Reply: Thank you for this suggestion. I will consider depositing the data in a repository. I have now stated that our study data is available upon request through our institute at this e-mail address: ipramho@kkh.com.sg

Reply: I have now matched the information provided in the ‘Funding Information’ and ‘Financial Disclosure’ sections. Please see Lines 774-775.

Lines 774-775: This study received funding from the National Medical Research Council (NMRC/CG/C008A/2017_KKH).

4. In the online submission form you indicate that your data is not available for proprietary reasons and have provided a contact point for accessing this data. Please note that your current contact point is a co-author on this manuscript. According to our Data Policy, the contact point must not be an author on the manuscript and must be an institutional contact, ideally not an individual. Please revise your data statement to a non-author institutional point of contact, such as a data access or ethics committee, and send this to us via return email. Please also include contact information for the third party organization, and please include the full citation of where the data can be found.

Reply: I have now provided an e-mail to contact which is the research team that manages the data. The e-mail has been provided in the manuscript. Please see Lines 769-770.

Lines 769-770: The datasets used and/or analysed during the current study are available from the IPRAMHO Research Team at ipramho@kkh.com.sg

Reply: The ethnics statements have now been moved to the Methods section of the manuscript in Lines 135-138.

Lines 135-138: The research procedures for this study received formal approval for an exempt review from the SingHealth Centralized Institutional Review Board (CIRB Ref No.: 2021/2610). All the authors and caregivers of the children surveyed have provided consent for the publication of this manuscript.

Reply: The captions for the Supporting Information have been listed at the end of the manuscript. The in-text citations have been updated and the supporting information files have been updated. Please see Lines 777-781.

Lines 777-781: 

S1 Table. Parental practices of lifestyle behaviours by age group 

S2 Table. Parental perceptions of child’s health by age group 

S3 Table. Parental awareness of lifestyle behaviour guideline recommendations by age group

Reply to reviewers' comments:

Reviewer 1

Reviewer #1: I believe this is an interesting study and there is a merit for this journal. Physical activity is an essential need for children. Unfortunately, sedentary life style and screen time are the most two important problems now. Authors made such a good systematic data collection and the date presented very well. However, reliability and validity of the tests and measurement should be reported. In addition, more up to date literature review (2020 and above) should be used in discussion section and references. In addition, future suggestions and practical applications should be added for future research studies. Finally, I wonder if authors checked any thing about gender and socio economic status of children or family income. These things could be important variables as well. Thank you.

Reply: Thank you for your comments and suggestions – I will be answering you queries in sections: 

“reliability and validity of the tests and measurement should be reported”

Reply: Firstly, I want to clarify that this study collected data through a survey questionnaire which was developed and reviewed by all the members of the Integrated 24-Hour Activity Guidelines for Early Childhood Committee. Thus, the development of the survey questionnaire had gone through an expert review and a thorough literature review which establishes its’ content validity. Furthermore, the questions used to capture physical activity, sleep and screen time have been adapted from validated questionnaires suitable for children of this age group which also justifies the validity and reliability of the outcome measures, especially for sleep and screen time viewing This has now been included in the revised version of the manuscript in Lines 149-155. The limitations of parent-reported data have been stated in the discussion of the manuscript, and ideally it would be beneficial to have a similar study repeated but collecting objectively measured data, especially for physical activity (Line 504-507). 

Lines 149-155: The survey questionnaire was developed and reviewed by members of the Integrated 24-Hour Activity Guidelines For Early Childhood Committee. Thus, the development of the survey questionnaire has gone through an expert review, and a thorough literature review which establishes content validity. Items on PA captured the total amount of tummy time for infants, total light and energetic play for toddlers and preschoolers using previously published questions [1], while the items on sleep were adapted from the Child Sleep Habits Questionnaire [2]. Recreational SVT was captured as the average time a child spends on a screen[3]. 

Lines 504-507: This study also has limitations, that need to be addressed. These include the subjective self-reported responses of caregivers in the survey about their children’s activities, which may not be as accurate as objective measurements using motion sensors.

“more up to date literature review (2020 and above) should be used in discussion section and references”

Reply: Thank you for this constructive suggestion. Since most of the studies published so far are in children of older age groups between 4-6 years, I have updated the references in this manuscript in Lines 413-420 and Lines 424-430 that discusses the results in pre-schooler aged children in this study. Four of these additional references are from year 2020 and above. 

Lines 413-420: In densely populated Asian cities like Hong Kong and Japan, only 2.9%[4] and 7.2% [5] respectively, of preschoolers were reported to comply with all three guidelines, a proportion lower than the rates observed in Singapore [6]. In Western populations, two studies revealed higher percentages of children adhering to all three guidelines [6, 7], while one observed very low rates of adherence [8] . In a study conducted in rural Brazil, 10% of children aged between 3-6 years met these criteria which concurred with our study observations [7]. In Australia, the proportion was slightly higher at 14.9% [6], whereas a study in the United States reported a percentage of 1.6% [8].

Lines 424-430: Again, similar to our findings, the study from Brazil has observed that approximately half of the children were meeting physical activity guidelines, while only approximately 20% were meeting screentime guidelines [7]. The trend of lower physical activity engagement during the weekdays concurs with the observations from the study conducted in Hong Kong [4], but contrasts with another study conducted in Singapore preschoolers, which reported higher engagements on weekdays, whether it was parent-reported data or objectively measured using wrist-worn accelerometers [9].

“future suggestions and practical applications should be added for future research studies.”

Reply: Thank you for this suggestion. I have now included in the Conclusion section more suggestions and practical applications for future research studies in Lines 514-520. Lifestyle interventions have traditionally focused exclusively on individual behaviors. However, future interventions may benefit from targeting integrated movement-related behaviors to enhance the adoption of healthy habits and address the issue of childhood obesity more comprehensively.

Lines 514-520: In conclusion, adherence to all three activity guidelines varied across age groups, showing a notable decline from infancy to preschool age. This decline was largely attributed to lower proportions of toddlers and preschoolers adhering to screen time guidelines compared to infants. Future interventions to address childhood obesity may benefit from targeting integrated movement-related behaviours, suggesting the need for strategies in Singapore to commence early, potentially through parenting education during pregnancy, to prevent the gradual decline in adherence across the years.

“gender and socio economic status of children or family income”.

Reply: Thank you for this suggestion. We have not explored potential risk factors that may be associated with high/low adherence to the 24-Hour Activity Guidelines like differences in gender or socio-economic status in this manuscript. But this is something we plan to explore separately in another analysis that will be published as a separate manuscript. 

Reviewer 2

Reviewer #2: Did that drop lead to worse results on some tests?

Or maybe to a worse state of health?

These would be useful answers on the basis of which it would be known whether it is justified to adhere to the 24-Hour Activity Guidelines.

This way the manuscript looks just like an overview newspaper article.

It has no scientific value.

By working with a control group, the manuscript would gain a completely new dimension. In this way, without a control group it is not scientific. It does not offer proofs and solutions.

I cannot recommend it for publication.

I think that it does not reach the quality level of the Plos One journal.

Reply: Thank you for your constructive comments on this manuscript. Yes, we do agree that more meaningful findings may be generated from the existing dataset. Presently, there is a paucity of publications addressing integrated activities in young children, especially within the Asian population, and no conclusive evidence specifically concentrates on children in Singapore. Any available evidence is still mostly in young children of the pre-schooler age, and there is descriptive data lacking in the infant and toddler age groups [10]. Therefore, the authors have reached a consensus to publish a descriptive manuscript encompassing various age groups, including infancy, toddlers, and preschoolers comparing the adherence to the 24-Hour Activity Guidelines across the age groups. 

We have previously published two other descriptive manuscripts on the 24-Hour Activity in Children and Adolescents in older age groups that were well received [11, 12].

Furthermore, this descriptive manuscript, being the inaugural publication from our study, will serve as a crucial reference to substantiate forthcoming planned interventions. Upon publication, this manuscript will serve as preliminary data for a study grant proposal, paving the way for a randomized control trial. The proposed trial seeks to examine children undergoing a parent-focused intervention aimed at fostering the adoption of the Integrated Activity Guidelines, comparing them with a control group. The trends revealed in this study, particularly the observed decline in adherence to all three guidelines as children enter the preschooler age group, emphasize the urgent necessity for interventions focused on parents. These interventions should aim to enhance all three activities in preschool-aged children.

With the current dataset we also plan to examine these hypotheses with the available data collected: 1) socio-demographic and caregiver predictors of adherence to 24-Hour Activity Guidelines in young children 2) Adherence to the 24-Hour Activity Guidelines and BMI in Young Children.

 

Reviewer 3

Reviewer #3: I read the study carefully and liked it very much. Methodologically, it is written according to the rules, I have only one dilemma that I leave to the editor to solve. Namely, in tables 1 and 2 there are missing data for a certain number of respondents (some of whom the author listed below the table, and some he failed to list), so the question arises whether some respondents who do not have complete data should be included in the study? Maybe the final results in some variables would be different? And on line 304, the data 71.9% do not agree with the data from table 4 (79.1%)

Reply: Thank you for highlight the error in the percentages presented. This has now been corrected in the manuscript in Lines 306-310:

Lines 306-310: On a weekend compared to a weekday, a higher proportion of toddlers met PA guidelines (68.9% vs 50.1%, p<0.001); conversely, a lower proportion met SVT guidelines (21.8% vs 38.8%, p<0.001), and there were no differences in meeting sleep guidelines (79.1% vs 82.6%, p>0.05). A higher proportion of toddlers met all three guidelines on weekdays, than on weekends (18.3% vs 13.2%, p<0.001).

In this descriptive analysis, we transparently present available data, including missing values, to offer an initial overview of patterns and trends. While recognizing this limitation, we intend to address missing data in future analyses, emphasizing the current analysis's value in providing a preliminary understanding of the subject.

For subsequent manuscripts that focus on specific hypotheses and involve more advanced statistical analyses to draw conclusions about relationships between variables, addressing missing data becomes crucial. For such analysis, we intend to exclusively utilize datasets devoid of any missing data points. Furthermore, to address missing data in any multivariable regression analysis, we plan to employ multiple imputation methods for robustness and accuracy.

 

References 

1. de Brito JN, Loth KA, Tate A, Berge JM. Associations Between Parent Self-Reported and Accelerometer-Measured Physical Activity and Sedentary Time in Children: Ecological Momentary Assessment Study. JMIR Mhealth Uhealth. 2020;8(5):e15458. doi: 10.2196/15458. PubMed PMID: 32348283; PubMed Central PMCID: PMCPMC7267997.

2. M. OJSAM. Children’s Sleep Habits Questionnaire (CSHQ): psychometric properties of a survey instrument for school-aged children. Sleep. 2000;23.

3. Bernard JY, Padmapriya N, Chen B, Cai S, Tan KH, Yap F, et al. Predictors of screen viewing time in young Singaporean children: the GUSTO cohort. Int J Behav Nutr Phys Act. 2017;14(1):112. doi: 10.1186/s12966-017-0562-3. PubMed PMID: 28870219; PubMed Central PMCID: PMCPMC5584344.

4. Feng J, Huang WY, Reilly JJ, Wong SH. Compliance with the WHO 24-h movement guidelines and associations with body weight status among preschool children in Hong Kong. Appl Physiol Nutr Metab. 2021;46(10):1273-8. Epub 20210504. doi: 10.1139/apnm-2020-1035. PubMed PMID: 33945770.

5. Tanaka C, Okada S, Takakura M, Hasimoto K, Mezawa H, Ando D, et al. Relationship between adherence to WHO “24-Hour Movement Guidelines for the Early Years” and motor skills or cognitive function in preschool children: SUNRISE pilot study. Japanese Journal of Physical Fitness and Sports Medicine. 2020 69(4):327-33. doi: https://doi.org/10.7600/jspfsm.69.327.

6. Cliff DP, McNeill J, Vella SA, Howard SJ, Santos R, Batterham M, et al. Adherence to 24-Hour Movement Guidelines for the Early Years and associations with social-cognitive development among Australian preschool children. BMC Public Health. 2017;17(Suppl 5):857. Epub 20171120. doi: 10.1186/s12889-017-4858-7. PubMed PMID: 29219104; PubMed Central PMCID: PMCPMC5773906.

7. Goncalves WSF, Byrne R, de Lira PIC, Viana MT, Trost SG. Adherence to 24-hour movement guidelines among rural Brazalian preschool children: associations with parenting practices. Int J Behav Nutr Phys Act. 2022;19(1):133. Epub 20221021. doi: 10.1186/s12966-022-01369-y. PubMed PMID: 36271449; PubMed Central PMCID: PMCPMC9587598.

8. McGowan AL, Gerde HK, Pfeiffer KA, Pontifex MB. Meeting 24-hour movement behavior guidelines in young children: Improved quantity estimation and self-regulation. Early Education and Development. 2022;34 (3):762-89. doi: https://doi.org/10.1080/10409289.2022.2056694.

9. Chen B, Waters CN, Compier T, Uijtdewilligen L, Petrunoff NA, Lim YW, et al. Understanding physical activity and sedentary behaviour among preschool-aged children in Singapore: a mixed-methods approach. BMJ Open. 2020;10(4):e030606. doi: 10.1136/bmjopen-2019-030606. PubMed PMID: 32265237; PubMed Central PMCID: PMCPMC7245386.

10. Rivera E, Hesketh KD, Orellana L, Taylor R, Carson V, Nicholson JM, et al. Prevalence of toddlers meeting 24-hour movement guidelines and associations with parental perceptions and practices. J Sci Med Sport. 2023. Epub 20231219. doi: 10.1016/j.jsams.2023.12.008. PubMed PMID: 38216403.

11. Quah PL, Loo BKG, Razali NS, Razali NS, Teo CC, Tan KH. Parental perception and guideline awareness of children's lifestyle behaviours at ages 5 to 14 in Singapore. Ann Acad Med Singap. 2021;50(9):695-702. doi: 10.47102/annals-acadmedsg.2021134. PubMed PMID: 34625757.

12. Quah PL, Loo BKG, Mettananda S, Dassanayake S, Chia MYH, Chua TBK, et al. 24 h Activity Guidelines in Children and Adolescents: A Prevalence Survey in Asia-Pacific Cities. Int J Environ Res Public Health. 2023;20(14). Epub 20230719. doi: 10.3390/ijerph20146403. PubMed PMID: 37510635; PubMed Central PMCID: PMCPMC10379132.

---

## [Decision Letter · Decision Letter 1]

2 Feb 2024

Adherence to 24-hour Integrated Activity Guidelines among Infants, Toddlers and Preschool children in Singapore.

PONE-D-23-34707R1

Dear Dr. Quah,

We’re pleased to inform you that your manuscript has been judged scientifically suitable for publication and will be formally accepted for publication once it meets all outstanding technical requirements.

Kind regards,

Stevo Popovic, Ph.D.

Academic Editor

PLOS ONE

Additional Editor Comments (optional):

Reviewers' comments:

Reviewer's Responses to Questions

**Comments to the Author**

1. If the authors have adequately addressed your comments raised in a previous round of review and you feel that this manuscript is now acceptable for publication, you may indicate that here to bypass the “Comments to the Author” section, enter your conflict of interest statement in the “Confidential to Editor” section, and submit your "Accept" recommendation.

Reviewer #1: All comments have been addressed

Reviewer #3: (No Response)

2. Is the manuscript technically sound, and do the data support the conclusions?

Reviewer #1: Yes

Reviewer #3: Yes

3. Has the statistical analysis been performed appropriately and rigorously? 

Reviewer #1: Yes

Reviewer #3: Yes

4. Have the authors made all data underlying the findings in their manuscript fully available?

Reviewer #1: Yes

Reviewer #3: Yes

5. Is the manuscript presented in an intelligible fashion and written in standard English?

Reviewer #1: Yes

Reviewer #3: Yes

6. Review Comments to the Author

Reviewer #1: I would like to thank to authors for revised version of this manuscript. The current version of this manuscript is acceptable. Best regards.

Reviewer #3: The authors have responded to my comments, and in my opinion, this manuscript is acceptable for publication in this respectable journal.

7. PLOS authors have the option to publish the peer review history of their article (what does this mean?). If published, this will include your full peer review and any attached files.

Reviewer #1: **Yes: **Ferman Konukman

Reviewer #3: **Yes: **Jovan Gardasevic

---

## [Editor Report · Acceptance letter]

16 Feb 2024

PONE-D-23-34707R1 

PLOS ONE

Dear Dr. Quah, 

I'm pleased to inform you that your manuscript has been deemed suitable for publication in PLOS ONE. Congratulations! Your manuscript is now being handed over to our production team.

Kind regards, 

on behalf of

Professor Stevo Popovic 

Academic Editor

PLOS ONE